

# Mechanical fault diagnosis of high voltage circuit breaker using multimodal data fusion

Tianhui Li[1,2], Yanwei Xia[1,2], Xianhai Pang[1,2], Jihong Zhu[3], Hui Fan[4], Li Zhen[4], Chaomin Gu[1,2], Chi Dong[1,2] and Shijie Lu[1,2]

[1] State Grid Hebei Electric Power Research Institute, Shijiazhuang, China
[2] State Grid Hebei Energy Technology Service Co., Ltd., Shijiazhuang, China
[3] Nanjing Hz Electric Co., Ltd., Nanjing, China
[4] State Grid Hebei Electric Power Supply Co., Ltd., Shijiazhuang, China

## ABSTRACT

A high voltage circuit breaker (HVCB) plays a crucial role in current smart power system. However, the current research on HVCB mainly focuses on the convenience and efficiency of mechanical structures, ignoring the aspect of their fault diagnosis. It is very important to ensure the circuit breaker conducts in a normal state. According to real statistics when HVCB works, most defects and faults in high voltage circuit breakers is caused by mechanical faults such as contact fault, mechanism seizure, bolt loosening, spring fatigue and so on. In this study, vibration sensors were placed at four different locations in the HVCB system to detect four common mechanical faults using vibration signal. In our approach, a convolutional attention network (CANet) was introduced to extract features and determine which mechanical faults occur within a fixed period of time. The results indicate that the mechanical fault diagnosis accuracy rate is up to 94.2%, surpassing traditional methods that rely solely on vibration signals from a single location.

## INTRODUCTION

The current smart power system puts forward the higher requirement of convenience, reliability and economy to its mechanical switch and operating mechanism. A common approach to monitoring the operational status of HVCB involves the installation of various sensors, including vibration, current measurement, and acoustic sensors. The data collected from these sensors not only indicate the occurrence of malfunctions but also help identify the types of faults. However, due to the limitations of using individual sensors and conflicts among data from different types of sensors, traditional methods exhibit several drawbacks. The cumbersome process of installing and dismantling sensors, coupled with their high sensitivity to operational conditions, renders the signals susceptible to interference. The variability in sensor types further exacerbates the reliability issues of the signals obtained. Additionally, diagnostic methods relying on a single source of signals may fail to accurately identify all faults, especially when the selected feature data derived from simulated environments do not encompass all potential fault scenarios. These

Corresponding author
Tianhui Li, tianhuili357@163.com

limitations significantly reduce the overall accuracy and reliability of such diagnostic approaches.

Most methods for diagnosing HVCB faults are based on vibration signals. *Qi, Gao & Huang (2020)* improved the efficiency of feature extraction and fault recognition for high voltage circuit breakers using a Light Gradient Boosting Machine based on time-domain feature extraction. *Chen & Wan (2021)* enhanced the performance of intelligent fault diagnosis for high-voltage circuit breakers by employing a Triangular Global Alignment Kernel Extreme Learning Machine (TGAK-ELM), resulting in more stable and accurate diagnostics. *Li et al. (2022)* introduced a multi-layer Integrated Extreme Learning Machine (IELM) for diagnosing mechanical faults in high-voltage circuit breakers, which significantly advanced diagnostic accuracy. *Zhang et al. (2022)* focused on the identification of mechanical faults in high voltage circuit breakers through multi-sensor information fusion, using wavelet packet decomposition and the Dempster-Shafer evidence theory. *Tahvilzadeh, Aliyari-Shoorehdeli & Razi-Kazemi (2023)* developed a model-aided intelligent fault detection system for SF6 high-voltage circuit breakers, utilizing simulation and machine learning algorithms to improve fault detection capabilities. *Yang et al. (2023)* optimized the fault diagnosis process for high voltage circuit breakers by implementing a Whale Optimization Algorithm-Support Vector Machine (WOA-SVM) based on principal component analysis (PCA), achieving greater diagnostic precision. *Cao et al. (2023)* created a method for localizing and identifying mechanical defects in high-voltage circuit breakers using vibration signal segmentation and chaotic feature extraction, effectively pinpointing common mechanical defects. *Chen et al. (2023)* introduced an ANFIS-based sound and vibration combined fault diagnosis method for high voltage circuit breakers, which significantly outperformed traditional methods in diagnostic accuracy. *Li et al. (2023)* proposed a robust fault diagnosis approach using an ensemble echo state network with evidence fusion, providing superior diagnostic performance through flexible and robust network parameters. *Liu et al. (2024)* suggested a defect diagnosis method for high voltage circuit breakers based on a combination of stroke curve and current signal, optimized by random forest, resulting in substantial improvements in diagnostic accuracy. *Xu et al. (2024)* implemented an intelligent mechanical fault diagnosis method employing Grey Wolf Optimization and Multi-Grained Cascade Forest algorithms, which enhanced the accuracy of diagnosing high-voltage circuit breakers.

In recent years, deep learning methods have provided effective solutions in pattern recognition and fault detection due to their powerful feature learning ability. *Ye et al. (2022)* proposed a novel U-Net with CapsNet for high-voltage circuit breaker fault diagnosis, achieving high precision and robust diagnosis in few-shot scenarios. This approach reduces feature loss during pooling and enhances diagnosis accuracy with a dynamic routing algorithm. Similarly, *Wang et al. (2022)* developed a few-shot transfer learning approach with an attention mechanism for high-voltage circuit breaker fault diagnosis, enhancing the robustness and accuracy of on-site diagnoses. This method leverages one-dimensional CNNs with attention mechanisms to focus on important parts of the fault signal. Furthermore, *Zhuang et al. (2022)* implemented a deep learning

approach for mechanical fault diagnosis of high voltage circuit breakers, utilizing time-frequency images of raw vibration data for high detection rates and low false alarm rates. This purely data-driven method surpasses traditional diagnosis models in performance. *Yan et al. (2023)* applied a Transformer-Convolutional Neural Network and Metric Meta-learning for few-shot mechanical fault diagnosis of high-voltage circuit breakers, achieving over 95% accuracy. This approach combines local and global feature extraction for robust fault classification. Additionally, *Zhang et al. (2023)* developed a PCA based Sparrow Search Algorithm (SSA) for optimizing a learning vector quantization (LVQ) neural network for mechanical fault diagnosis of high voltage circuit breakers, enhancing the diagnostic rate and training convergence. This method reduces feature redundancy and improves recognition accuracy. Moreover, *Wang et al. (2023)* proposed a hybrid transfer learning approach for on-site small-sample high-voltage circuit breaker fault diagnosis, combining domain adaptation and domain adversarial training to achieve high accuracy. This method ensures that the network focuses on key features while fully extracting temporal information. Similarly, *Zheng et al. (2023)* proposed a prediction method for the mechanical state of high-voltage circuit breakers based on long short-term memory (LSTM) neural network and support vector machine (SVM), enabling predictive maintenance and enhancing system reliability. This method predicts key mechanical parameters and diagnoses the mechanical state using predicted data. Finally, *Sui et al. (2024)* proposed a Dynamic Multi-Attention Graph Convolutional Network (DMGCN) for mechanical fault diagnosis of high-voltage circuit breakers, utilizing adaptive graph construction for better performance. This method effectively integrates structural and numerical information for improved classification.

In our work, we have developed a data fusion technique aimed at overcoming the limitations of traditional fault diagnosis methods by using multiple signals of the same type. Considering that vibration signals are easy to acquire and analyze, our proposed approach collects vibration signals from four distinct locations on the HVCB to identify four types of mechanical faults: contact faults, mechanism seizures, bolt loosening, and spring fatigue.

This article employs deep learning approach to analyze vibration signals from different positions, aiming to detect faults during the operation of HVCB. Initially, data on the normal operation and mechanical faults of the high-voltage circuit breaker (HVCB) are collected from different locations where four vibration sensors are installed. This data is used to construct a dataset. Subsequently, deep learning techniques are employed to detect mechanical faults. All fault signals are generated from real operational environments. Finally, we established a convolutional attention network (CANet) to extract signal features for classifying mechanical faults.

Our contributions are summarized as follows:

(1) A high voltage circuit breaker fault detection platform was constructed, where vibration signals collected from four different locations on the breaker effectively identify contact faults, mechanism seizures, bolt loosening, and spring fatigue.

(2) A simple and effective CANet was developed, utilizing the combination of one-dimensional convolution and self-attention mechanisms to efficiently extract key features from multi-point time series data and classify faults.

(3) Multi-point data fusion was implemented, combining vibration signals from various locations on the high voltage circuit breaker, overcoming the limitations of traditional fault diagnosis methods that rely on a single signal source, and enhancing the accuracy and reliability of fault diagnosis.

The remainder of the article is organized as follows. We describe the proposed method in "Materials and Methods". The performance of the CANet is evaluated in "Result". Finally, we provide discussion and conclusions in "Discussion" and "Conclusion".

## MATERIALS AND METHODS

In this article, we focus on a specific 12kV high-voltage circuit breaker with a spring operating mechanism as our subject. Our primary objective was to detect the four most common faults in HVCBs: contact fault, mechanism seizure, bolt loosening and spring fatigue. The signals were downsampled and concatenated, and fault diagnosis was conducted on the combined data using our CANet, ultimately determining the presence of faults.

### Signal acquisition

This study focuses on the 12kV spring-operated high-voltage circuit breaker model ZW32-12F. Vibration signals at different operating positions of the spring-operated mechanism were collected using TX9R033-2 vibration sensors. As shown in Fig. 1, the vibration sensors were installed at various positions on the operating mechanism and the chassis using strong magnets attached to the bottom of the sensors. By deploying multiple vibration sensors, vibration signals from different positions on the operating mechanism and the chassis were obtained. Additionally, acoustic sensors were used to collect sound signals during the sampling process, although these signals were not used for fault diagnosis. The collected signals were transmitted to a data storage unit *via* an NI 9234 data acquisition card, with the sampling frequency controlled by edge computing devices set at 25.6 kHz. Finally, the collected data were uploaded to a cloud server. This study captures the vibration signals of the operating mechanism and the chassis during HVCB disconnection by setting a threshold to determine the start time of the vibration signals.

### Convolutional attention network

To effectively detect and classify faults in high-voltage circuit breakers, our approach involves a comprehensive analysis of collected data from various locations. These signals are pivotal in determining the operational state of HVCB over time. To achieve this, we have developed a specialized neural network model, termed CANet, designed to extract and process features from these varied inputs and ultimately categorize the different types of faults using a multi-layer perceptron (MLP).

The methodology begins with the selection of vibration signals of a fixed length, ensuring a uniform input structure. These signals are then downsampled at a

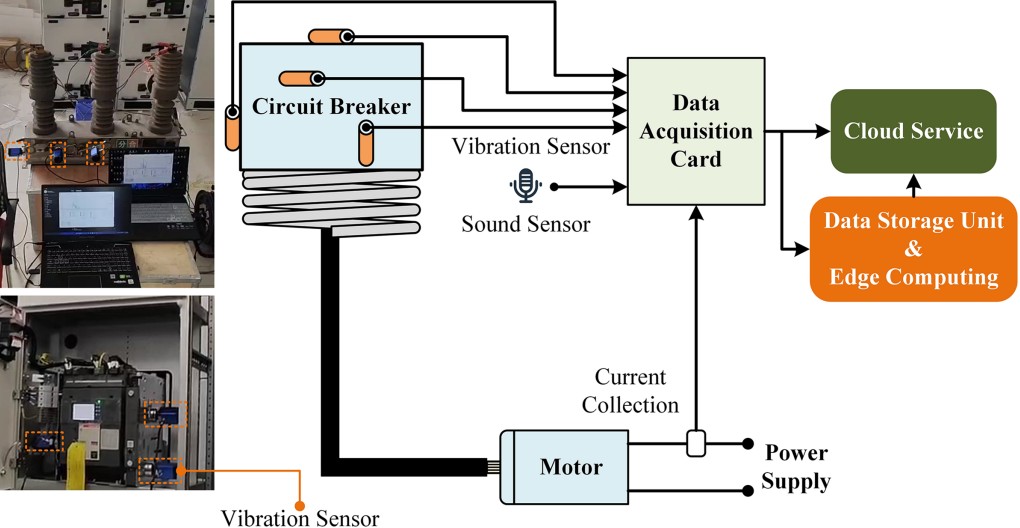

**Figure 1 The vibration, sound and current signal acquisition.**

predetermined frequency to standardize the data and reduce noise, making them suitable for deep learning processing. The downsampled signals are then concatenated, creating a composite signal that combines the characteristics of each modality. This composite signal is fed into the CANet for in-depth feature extraction.

CANet primarily consists of a one-dimensional convolution module and an attention module. In the one-dimensional convolution process, filters slide along the time series, engaging in point-wise multiplication with different parts of the series to effectively capture local dependencies and features. By employing various filters, the one-dimensional convolution can extract a diverse range of features from the raw time series. The attention mechanism enables the model to focus more on the parts that are most critical for the current task while processing information. When handling time series data, the attention mechanism is particularly useful in identifying the information most relevant to the current task. Compared to traditional convolutional networks, the attention mechanism offers a more dynamic way of extracting features, as it can adaptively adjust its focus on different parts of the series based on the task requirements and contextual changes.

The architecture of CANet is designed to handle the complexity of multi-point data. The input tensor $\in R^{L \times 4}$, where L represents the signal length, is first subjected to one-dimensional convolution. This process is pivotal for extracting and expanding features from the vibration data. The convolutional layer not only captures the inherent characteristics of each signal type but also begins the process of blending these features to form a comprehensive fault signature. The vibration signals after feature extraction are shown in Fig. 2.

Following initial feature extraction, the data are fed into the convolutional attention (CA) module for further processing. This module employs both one-dimensional convolution and attention modules to extract spatial and temporal features, offering a more nuanced understanding of the data. The convolutional component of the CA module

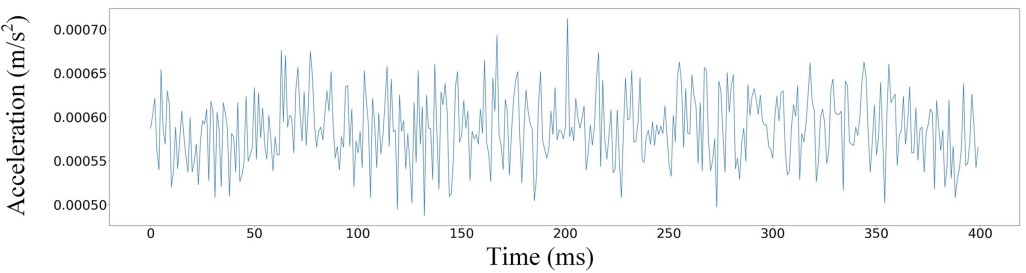

(a) The signal from vibration sensor A.

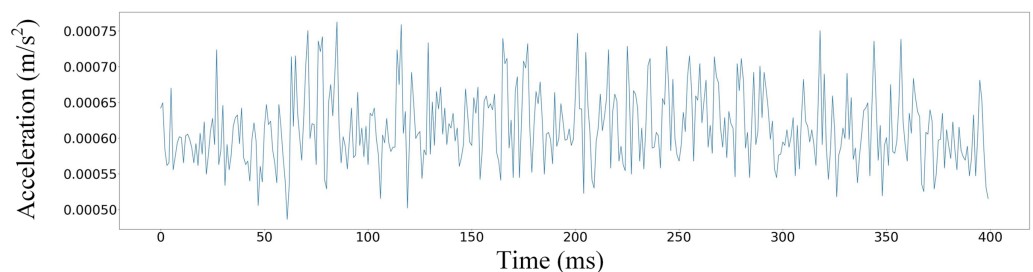

(b) The signal from vibration sensor B.

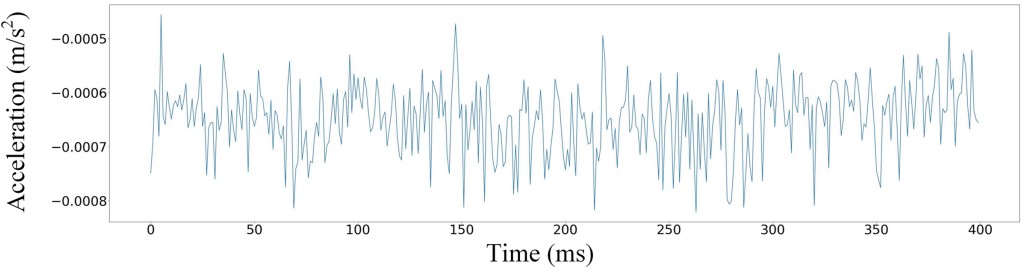

(c) The signal from vibration sensor C.

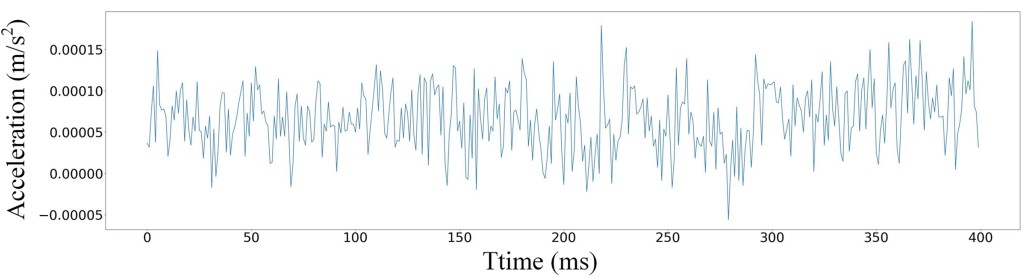

(d) The signal from vibration sensor D.

**Figure 2** **Partial signals collected by vibration sensors (A–D) after feature extraction.**

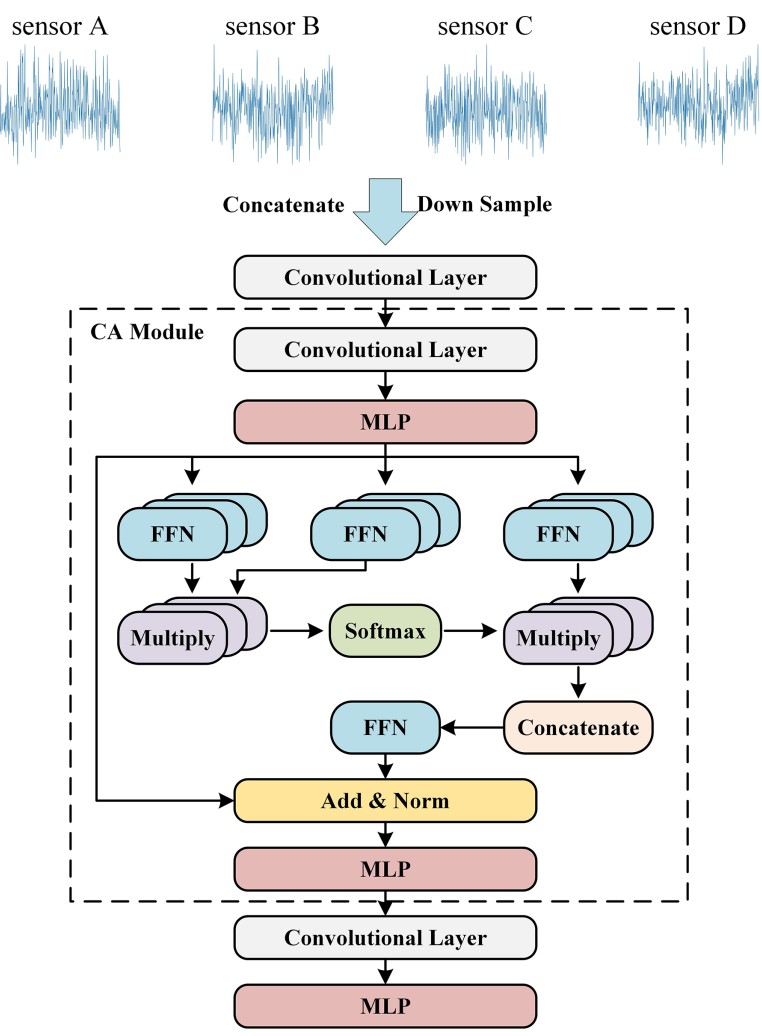

**Figure 3 The architecture of the algorithm.**

works to fuse features across different channels, effectively synthesizing information from vibration signals collected at various locations. The attention module then comes into play, focusing on crucial segments of the data sequence. It identifies patterns and correlations over time, bringing out subtle but significant anomalies indicative of potential faults.

Finally, the processed data are passed through a final one-dimensional convolutional layer. This layer integrates the temporal information and prepares the data for fault classification. The MLP module then takes over, applying its powerful classification capabilities to accurately categorize the faults in HVCB.

## CA module

As shown in Fig. 3, the CA module we designed utilizes one-dimensional convolutional modules and attention modules to effectively extract features from input data across temporal and spatial dimensions, enhancing the interaction between local and global features. Initially, the one-dimensional convolutional module captures local features of the time-series signal, expanding the feature space to facilitate the exchange of information

across channels. After convolution, the MLP is used to further enrich the feature tensor, introducing non-linearity to uncover more complex data characteristics. Following this, the application of the attention module allows us to focus on crucial parts of the sequence, capturing global correlations within the time-series signal.

We employ a multi-head attention module to adaptively learn different features. The tensor is passed through three separate FFN modules, generating Query, Key, and Value respectively. We first multiply Query and Key, and then apply a Softmax function to generate the corresponding heatmap. Following this, we multiply the heatmap with Value. Finally, we concatenate the results along the channels and project them through an FFN for mapping. The specific expression formula is as follows:

$$MultiHead(Q, K, V) = Concat(head_1, \ldots, head_h)W^o$$
$$head_i = Attention(QW_i^Q, KW_i^K, VW_i^V)$$
$$Attention(Q_i, K_i, V_i) = softmax\left(\frac{Q_i K_i^T}{\sqrt{d_k}}\right)V_i$$

(1)

where $h$ denotes the number of heads, $head_i$ represents the output of the $i^{th}$ head, and $W^o$ is the output transformation matrix. $W_i^Q$, $W_i^K$, $W_i^V$ are the transformation matrices for query, key, and value, respectively, and $d_k$ is the dimensionality of the key tensor.

Ultimately, the features outputted by the convolutional layer are combined with those enhanced by the attention module through residual connections. This not only preserves the original information but also enhances the expressiveness of features, providing robust support for deep learning and fault detection within the model.

## RESULTS

### Implementation details

Following the installation method described in 'Signal Acquisition', we collected a total of 42,468 data entries. A total of 200 vibration signals were collected during the high-voltage circuit breaker operation. Among them, there were 17 sets of vibration signals with contact faults, nine sets with mechanism seizure, 23 sets with bolt loosening, and 10 sets with spring fatigue. Due to the limited number of faulty vibration signal data, it is necessary to augment the data. The method involves overlaying faulty vibration signals with normal vibration signals and adding random noise to generate a completely new faulty vibration signal. Similarly, multiple faulty signals can be overlaid to augment the dataset. To test the performance of our models, we employed the Monte Carlo cross-validation method. We randomly selected 80% of the continuous data from the entire dataset and split it into a training set and a test set in a 4:1 ratio. Each model was trained five times following the aforementioned method, and we saved the weights of the model that performed best on the test set. The average accuracy of each model was calculated and used as a key indicator to measure model performance.

The entire process was conducted in a Python environment, using the PyTorch framework for building, training, and testing the models. We only employed one CA modules to construct CANet. The output dimension of the first convolutional layer is set to

128, with a kernel length of five. In the CA module, the input and output feature dimensions of the first convolution operation are both 128, with a kernel length of five. The MLP is configured with a three-layer structure, where the input and output dimensions are both 128, and the hidden layer dimension is 256. The number of heads in the attention module is eight. In the final convolutional layer, the input dimension is 128, and the output dimension is 256, with a kernel length of 50. The ultimate MLP is also configured with a three-layer structure, where the input dimension is 256, the output dimension is four, and the hidden layer dimension is 512.

During the training process, we trained the model for 100 epochs, set the learning rate to 1e-4, and fixed the batch size at 256. We set the length of the time series data to 50 and used a GTX 2080Ti graphics card for training the models.

## Evaluation metrics

To evaluate the performance of our constructed CANet in detecting four types of faults in high-voltage circuit breakers, we utilize accuracy, precision, recall, and the F1-score (F1 measure) as metrics to measure the performance of CANet on our custom-built dataset. The following sections will provide a detailed introduction to these evaluation metrics.

Accuracy refers to the proportion of samples that are correctly classified out of the total number of samples. Precision and Recall are the most commonly used metrics in evaluating classification tasks. Precision is defined as the proportion of true positive samples among all samples predicted as positive. Recall, from the perspective of the original samples, is defined as the probability of samples that are truly positive being predicted as positive. The mathematical definitions of these metrics are as follows:

$$
\begin{aligned}
Precision &= \frac{True\ Positive}{True\ Positive + False\ Positive} \\
Recall &= \frac{True\ Positive}{True\ Positive + False\ Negative}.
\end{aligned}
\tag{2}
$$

To take into account both Precision and Recall, the F1-score is commonly used as a measure. The F1-score is the harmonic mean of Precision and Recall, designed to balance the two metrics. It is defined as:

$$
F1 = 2 \cdot \frac{Precision \cdot Recall}{Precision + Recall}.
\tag{3}
$$

## Experiments

In our dataset, we randomly allocated 80% of the data for training and testing purposes, and conducted fivefold cross-validation. The CANet produces an output tensor of length four, representing four types of faults: contact fault, mechanism seizure, bolt loosening and spring fatigue. In this study, we adapted existing models—LSTM, GRU, TIMESNet (*Wu et al., 2022*) and Anomaly Transformer (*Xu et al., 2022*)—by modifying their source code to accept our data for fault detection. We then conducted a comparative analysis across these models and CANet, evaluating their performance on four types of mechanical faults

**Table 1** Performance comparison of CANet, LSTM, GRU, TIMESNET, and Anomaly Transformer in mechanical fault detection of HVCB (The maximum value for each metric is highlighted in bold).

| Fault | Contact fault | | | Mechanism seizure | | | Bolt loosening | | | Spring fatigue | | | Average | | | |
|---|---|---|---|---|---|---|---|---|---|---|---|---|---|---|---|---|
| Metric | P | R | F1 | P | R | F1 | P | R | F1 | P | R | F1 | A | R | P | F1 |
| LSTM | 0.939 | 0.881 | 0.909 | 0.915 | 0.921 | 0.918 | 0.918 | 0.891 | 0.904 | 0.933 | 0.891 | 0.911 | 0.841 | 0.894 | 0.927 | 0.911 |
| GRU | 0.924 | 0.923 | 0.924 | 0.951 | 0.946 | 0.948 | 0.931 | 0.911 | 0.921 | 0.938 | 0.919 | 0.929 | 0.873 | 0.924 | 0.935 | 0.929 |
| TIMESNET | 0.952 | 0.975 | 0.963 | 0.957 | 0.936 | 0.947 | 0.926 | 0.963 | 0.944 | 0.938 | 0.956 | 0.947 | 0.909 | 0.959 | 0.943 | 0.951 |
| Anomaly transformer | 0.964 | **0.992** | **0.978** | **0.994** | 0.882 | 0.932 | **0.976** | 0.978 | **0.976** | **0.994** | **0.996** | **0.994** | 0.934 | 0.949 | **0.984** | 0.966 |
| CANet(Ours) | **0.974** | 0.978 | 0.976 | 0.964 | **0.948** | **0.954** | 0.958 | **0.982** | 0.97 | 0.992 | 0.992 | 0.992 | **0.941** | **0.972** | 0.97 | **0.97** |

**Table 2** Performance of CANet on the dataset.

| No. | Accuracy | Recall | Precision | F1-score |
|---|---|---|---|---|
| I | 0.965 | 0.986 | 0.987 | 0.986 |
| II | 0.938 | 0.982 | 0.961 | 0.971 |
| III | 0.937 | 0.954 | 0.976 | 0.965 |
| IV | 0.926 | 0.954 | 0.961 | 0.957 |
| V | 0.941 | 0.984 | 0.967 | 0.975 |
| Ave. | 0.942 | 0.972 | 0.970 | 0.971 |

in HVCB in terms of precision, recall, and F1-scores. The result is as shown in Table 1. The results demonstrate that CANet slightly outperforms Anomaly Transformer in terms of overall recall and F1-score, indicating a superior ability to detect all relevant cases and maintain a balance between precision and recall. While Anomaly Transformer exhibits a marginally higher average precision, suggesting fewer false positives, the differences between the two methods are relatively minor.

The classification results of CANet are presented in Table 2. In Table 2, we present the accuracy, recall, precision, and F1-score of CANet across fivefold cross-validation. Averaging the results of the fivefold cross-validation, we obtained an accuracy of 0.942, a recall of 0.972, a precision of 0.970, and an F1-score of 0.971. The high accuracy indicates that CANet performs well across the entire dataset. The high recall and precision demonstrate that the model effectively identifies and classifies positive samples with minimal misclassification of negative samples as positive. An F1-score of 0.971 reflects a good balance between precision and recall, showcasing the model's excellent overall performance.

In Table 3, we present the average detection performance metrics of CANet for four types of faults under fivefold cross-validation. The faults include contact fault, mechanism seizure, bolt loosening and spring fatigue. We list the Precision, Recall, and F1-score for the detection of these four faults. Among these faults, CANet demonstrates the best detection performance for spring fatigue, with Precision, Recall, and F1-score all approaching one. The detection performance for the other three faults is also high, with Precision and Recall exceeding 0.95, and F1-scores close to or exceeding 0.95.

**Table 3 Four types of fault detection performance.**

| Types of faults | Precision | Recall | F1-score |
|---|---|---|---|
| Contact fault | 0.974 | 0.978 | 0.976 |
| Mechanism seizure | 0.964 | 0.948 | 0.954 |
| Bolt loosening | 0.958 | 0.982 | 0.97 |
| Spring fatigue | 0.992 | 0.992 | 0.992 |

**Table 4 Five-fold cross-validation F1-scores for four types of fault.**

| No. | C. F1-score | M. F1-score | B. F1-score | S. F1-score |
|---|---|---|---|---|
| I | 0.97 | 1.0 | 0.98 | 0.99 |
| II | 0.92 | 0.98 | 0.95 | 0.99 |
| III | 1.0 | 0.94 | 0.96 | 1.0 |
| IV | 1.0 | 0.91 | 0.97 | 0.99 |
| V | 0.99 | 0.94 | 0.99 | 0.99 |
| Ave. | 0.976 | 0.954 | 0.97 | 0.992 |

**Table 5 Five-fold cross-validation precision for four types of fault.**

| No. | C. Precision | M. Precision | B. Precision | S. Precision |
|---|---|---|---|---|
| I | 0.95 | 1.0 | 0.99 | 0.99 |
| II | 0.93 | 0.98 | 0.90 | 0.99 |
| III | 1.0 | 0.98 | 0.95 | 1.0 |
| IV | 1.0 | 0.94 | 0.95 | 1.0 |
| V | 0.99 | 0.92 | 1.0 | 0.98 |
| Ave. | 0.974 | 0.964 | 0.958 | 0.992 |

In Table 4, we have listed the F1-scores for four types of faults in each round of the fivefold cross-validation. "C. F1-score" represents the F1-score in detecting contact fault, "M. F1-score" for mechanism seizure, "B. F1-score" for bolt loosening, and "S. F1-score" for spring fatigue. The F1-scores for all four types of fault detection are generally very high, indicating that the model possesses a high level of accuracy and reliability. The model's performance is relatively stable across different rounds, with only minor fluctuations occurring in a few instances. However, these fluctuations are minimal, demonstrating the model's robustness in detecting various types of faults. For the mechanism seizure, the F1-scores are slightly lower in some rounds (dropping to as low as 0.91), which could suggest that this type of fault is more challenging to detect compared to others, or that the dataset features for this part are not sufficiently distinct.

In Tables 5 and 6, we have listed the precision and recall for four types of faults in each round of the fivefold cross-validation. CANet achieves a Precision and Recall of over 0.95 in detecting these four types of faults. This indicates that CANet is both accurate and

**Table 6 Five-fold cross-validation recall for four types of fault.**

| No. | C. Recall | M. Recall | B. Recall | S. Recall |
|---|---|---|---|---|
| I | 0.99 | 0.99 | 0.97 | 0.99 |
| II | 0.92 | 0.98 | 0.99 | 0.99 |
| III | 1.0 | 0.91 | 0.97 | 0.99 |
| IV | 1.0 | 0.89 | 0.99 | 0.99 |
| V | 0.98 | 0.97 | 0.99 | 1.0 |
| Ave. | 0.978 | 0.948 | 0.982 | 0.992 |

comprehensive in its fault detection capabilities, particularly excelling in the detection of spring fatigue. The minor variations in Precision and Recall across different data splits suggest that the model's performance is stable and not significantly affected by specific data partitions, demonstrating good robustness.

## DISCUSSION

Our study employs multi-point data fusion technology for diagnosing mechanical faults in HVCB. Traditional HVCB fault diagnosis methods primarily focus on the convenience and efficiency of mechanical structures, often overlooking the crucial aspect of fault diagnosis. Our research enhances fault detection accuracy and reliability by aggregating multi-point vibration signals.

Traditional fault diagnosis techniques usually rely on a single signal source or manual feature selection, limiting their diagnostic capabilities. To address this, we introduced the deep learning model CANet. CANet is capable of processing multi-point data, providing a more comprehensive analysis of the circuit breaker's condition. This is particularly evident in its superior performance in detecting faults such as contact fault and spring fatigue.

One of our main findings is the effectiveness of the attention mechanism in CANet. The inclusion of the attention mechanism not only improved the average accuracy but also enhanced the model's ability to detect each type of fault. This underscores the value of integrating such mechanisms into deep learning models, especially for applications involving time-series data.

While CANet demonstrates excellent fault detection performance, it is less satisfactory in identifying certain faults, such as mechanism seizure and bolt loosening. This suggests a need for further refinement of the model, possibly by expanding the dataset to include a wider range of fault types and operational environments. Such enhancements could improve the model's generalizability and robustness.

Transformers have great potential in the fault diagnosis process. Known for their effectiveness in handling sequence data, transformers could further enhance fault detection accuracy. This aligns with the broader trend in machine learning and deep learning research, where the focus is shifting towards models that can effectively process and analyze complex data structures. In addition, the integration of transformer models into fault diagnosis systems in real-world settings could be challenging due to their high computational demands, requiring substantial hardware resources. Future studies might

explore the development of more efficient transformer models or hybrid approaches that combine the strengths of different models.

Our study demonstrates the substantial potential of combining multi-point data fusion with attention mechanisms in the field of HVCB mechanical fault diagnosis. We plan to explore the use of improved transformer models to further increase fault diagnosis accuracy and to expand the dataset to include a wider range of fault types and various operational environments.

Future research directions could include exploring methods to mitigate the impact of environmental variables on signal accuracy and developing more cost-effective deployment strategies. Additionally, integrating advanced data preprocessing techniques to handle diverse environmental conditions could be a key area of focus.

## CONCLUSIONS

This article establishes a mechanical fault diagnosis method for HVCB based on multi-point data fusion technology. By integrating multi-point data fusion, we have successfully addressed the limitations of traditional fault diagnosis methods, which primarily focus on mechanical structure efficiency and often neglect the crucial aspect of accurate fault detection. Our work can be summarized as follows:

(1) We constructed a platform for collecting mechanical fault data of HVCBs using vibration sensors. By using fault injection methods to expand the dataset, we have overcome the challenges associated with the scarcity of fault data.

(2) We developed a simple yet effective CANet, utilizing one-dimensional convolution and an attention mechanism for feature extraction from multi-point data. The capability of CANet to process and analyze multi-point data has been proven to be highly effective.

(3) We have validated the capability of the attention mechanism in processing multi-point time-series data. Our findings demonstrate that the attention mechanism is a key factor in enhancing the diagnostic capabilities of CANet, not only improving overall accuracy but also increasing the precision of the model in identifying various types of faults.

Overall, our research not only proves the effectiveness of combining deep learning with multi-point data fusion for mechanical fault diagnosis in HVCBs but also lays the groundwork for further innovations in the important field of smart power systems. In the future, we will continue to explore advanced deep learning techniques, aiming to provide more efficient, accurate, and reliable fault diagnosis methods for high voltage circuit breakers.

### Funding

This work was supported by the State Grid Hebei Electric Power Co., Ltd. Technology Project Funding (kj2022-062) and the National Natural Science Foundation of China (No. 62371253 and No. 52278119). The funders had no role in study design, data collection and analysis, decision to publish, or preparation of the manuscript.

## Grant Disclosures

The following grant information was disclosed by the authors:
State Grid Hebei Electric Power Co., Ltd. Technology Project Funding: kj2022-062.
National Natural Science Foundation of China: 62371253 and 52278119.

## Competing Interests

Tianhui Li, Yanwei Xia, Xianhai Pang, Chaomin Gu, Chi Dong and Shijie Lu are employed by State Grid Hebei Electric Power Research Institute and State Grid Hebei Energy Technology Service Co., Ltd. Jihong Zhu is employed by Design and Development Department, Nanjing Hz Electric Co., Ltd. Hui Fan and Li Zhen are employed by State Grid Hebei Electric Power Supply Co., Ltd. The authors declare that they have no competing interests.

## Author Contributions

- Tianhui Li conceived and designed the experiments, authored or reviewed drafts of the article, and approved the final draft.
- Yanwei Xia conceived and designed the experiments, authored or reviewed drafts of the article, and approved the final draft.
- Xianhai Pang conceived and designed the experiments, analyzed the data, authored or reviewed drafts of the article, and approved the final draft.
- Jihong Zhu conceived and designed the experiments, analyzed the data, authored or reviewed drafts of the article, and approved the final draft.
- Hui Fan conceived and designed the experiments, analyzed the data, performed the computation work, prepared figures and/or tables, and approved the final draft.
- Li Zhen analyzed the data, performed the computation work, prepared figures and/or tables, and approved the final draft.
- Chaomin Gu performed the experiments, analyzed the data, performed the computation work, prepared figures and/or tables, and approved the final draft.
- Chi Dong performed the experiments, performed the computation work, prepared figures and/or tables, and approved the final draft.
- Shijie Lu performed the experiments, prepared figures and/or tables, and approved the final draft.

## Data Availability

    The code is available in the Supplemental Files.
    The data and code are available at GitHub and Zenodo:
    - https://github.com/tianhuiLi700/CAnet.
    - https://doi.org/10.5281/zenodo.13291531.

## Supplemental Information

Supplemental information for this article can be found online at http://dx.doi.org/10.7717/peerj-cs.2248#supplemental-information.

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
