# Peer review of "Mechanical fault diagnosis of high voltage circuit breaker using multimodal data fusion"

_PeerJ Computer Science, doi:10.7717/peerj-cs.2248_

## Round 0.1 · original submission · Major Revisions

Dear Authors,

Your paper has been revised. Based on the reviewers' analysis the paper needs of major revisions before being considered for publication in PeerJ Computer Science.

More precisely, despite the manuscript's very interesting and well-structured topic, the introduction and experimental parts of the paper must be improved.

Regarding the introduction part, the authors must rewrite it, adding more research literature on high-voltage circuit breakers.

About the experimental part, the experimental data collection and experimental setup are weakly described. Furthermore, the authors must address the following questions:

1) What is the manufacturer and exact model of the HVCB used in the experiment? This information should be included due to various breaking chamber designs, operating mechanism implementations, and differences in other subassemblies of the HVCB, even for the same rated voltage.

2) The accelerometer mounting approach should be clarified. Several mounting approaches are used in published academic manuscripts and industrial applications: fix mounting by bolts, glue, strong magnets, or beeswax. All mentioned approaches yield differences in the frequency representation of the signal. Which one has been utilized in the experiment?

3) For the method to be reproducible, there should be more details regarding the DL network.

**Language Note:** The review process has identified that the English language must be improved. PeerJ can provide language editing services - please contact us at [email protected] for pricing (be sure to provide your manuscript number and title). Alternatively, you should make your own arrangements to improve the language quality and provide details in your response letter. – PeerJ Staff

Reviewer 1 ·

Basic reporting

The topic of the manuscript is very interesting and well-structured. However, there is a lot of room for improvement, especially in terms of writing and experimental parts. Here are some comments and suggestions:

1. Language and writing:

There are a lot of typos and sentences that might be written more concisely. There is a lot of room for writing improvement. Here are some of the comments for the improvement of the manuscript writing:

1.1. Abstract
- Line 17, Missing the article: „High voltage circuit breaker (HVCB) plays a crucial role in the current smart power system.“
- Line 23: It is stated that the sensors are mounted in the „HVAC system“? I assume that HVAC should be the HVCB?

1.2. Introduction
Line 45: Correct the sentence as: „In recent years, Deep learning methods have provided effective solutions in pattern recognition and fault detection due to their powerful feature learning ability.“

1.3. Results
- Line 205 – „Following the installation method described in Section 2.1....“. Previous sections are not numbered thus it is hard to follow which section is Section 2.1.

Figure 1. Typo "Cloud sevice" -> "Cloud service"

Those are not the only typos and writing issues.

1.3. Field background:
Most of the surveyed papers are not related to the fault diagnostics of the HVCB but to other objects such as PV plants, wind turbines, bearings etc. If the surveyed paper covers the field of assessment and fault detection or isolation of the HVCB in any way, it should be stated and elaborated.

Experimental design

The experimental data collection and experimental setup are weakly described. A lot of unanswered questions arise while reading the manuscript. The Authors should address the following questions:

1. Experimental setup:

- What is the manufacturer and exact model of the HVCB used in the experiment? Due to various breaking chamber designs, operating mechanism implementations and differences in other subassemblies of the HVCB, even for the same rated voltage, this information should be included.
- The position of the accelerometer should be precisely depicted on the schematic of the HVCB. Whether the accelerometers are mounted on the chassis or operating mechanism? The position of the accelerometer highly affects the obtained data and the classification performance.
- The accelerometer mounting approach should be clarified. There are several mounting approaches used throughout published academic manuscripts and industrial applications: fix mounting by bolts, glue, strong magnets or beeswax. All mentioned approaches yield differences in the frequency representation of the signal. Which one has been utilized in the experiment?

2. Collecting and preprocessing data:
- I assume that the data have been collected during the making (closing) or breaking (opening) operation of the HVCB. That is not clearly stated anywhere in the manuscript. Also, this is not reflected in Figure 2. This is crucial.
- It is stated that all signals are acquired using the same sampling rate. (Line 144 and 145). What is the exact sampling rate used in this experiment? What are the criteria for the sampling rate selection?
- Line 153. „Signals are downsampled at a predetermined frequency...“. What are the criteria for the downsampling rate selection?
- How the start time of the measurement is defined? Is the measurement triggered on the coil current threshold, digital command impulse to the HVCB to operate or something else?

3. DL network
In order for the method to be reproducible, there should be more details regarding the DL network. You have two convolution layers but there are no data such as kernel filter size. The rest of the network should be precisely defined in the manuscript. This is clear from supplementary materials, but it should be included in the manuscript as well.

4. Definitions

Validity of the findings

In general, the concept of the proposed topic is well-structured. However, without details such as experimental setup details as well as details of the deep learning network (number of filters, etc...). The method proposed in the manuscript must be reproducible for another dataset. Those details degrade the achieved algorithm performance.

Nevertheless, the achieved performance is very well and promising.

Reviewer 2 ·

Basic reporting

The paper was not formatted according to the standard format.

Experimental design

The experimental design is too simple, lacking rationality and comprehensiveness.

Validity of the findings

No

Additional comments

1.The principle and equipment for collecting the voiceprint signal and vibration signal of the circuit breaker are different. Figure 1 presented in the article only uses one sensor to obtain them, which raises doubts about its implementation process.
2. The acquisition of voiceprint signals and vibration signals is not only solved by a single sensor, but also requires corresponding supporting devices, such as designed circuits, storage devices, power supplies, etc., which were not explained clearly in the article.
3. The convolutional neural network algorithm was used in the article to extract features from signals, but the method provided was not explained clearly. How to apply algorithms for feature extraction is not explained in detail in the article.

Reviewer 3 ·

Basic reporting

The paper proposes a mechanical fault diagnosis method for high-voltage circuit breakers (HVCBs) based on multimodal data fusion. This method collects vibration signals from different locations and uses deep learning techniques to detect and classify typical mechanical faults of HVCBs. The paper introduces a Convolutional Attention Network (CANet) for feature extraction from vibration signals. By installing vibration sensors at various locations to collect data, a dataset was constructed, and deep learning technology was employed for fault detection. Ultimately, the effectiveness of the method was verified through experiments. The paper has the following problems that require further revision.

1. Fault diagnosis methods based on deep learning are widely introduced in the introduction. However, there is almost no literature on high voltage circuit breakers. The authors need to restructure the introduction to add research literature on high voltage circuit breakers.
2. The pictures in the paper are not standardized. For example, Figure 2 is missing axis names and units.
3. The formatting of formulas in the paper is not uniform.

Experimental design

There are some problems with the experiment that require further explanation and clarification by the authors.
1.There are more details in the paper that are not explained. The authors artificially inserted the fault, however, the paper lacks detailed information about the fault. How are the faults modeled? Are there vibration signals modeled for different faults?
2.The paper lacks a detailed description of the training and testing samples.

Validity of the findings

The results of the experiment are valid, but some details of the methods need to be added.
1.In deep learning methods, the detailed structure and parameters of the network have a great impact on its performance. The paper lacks detailed information in this regard. It is suggested that the authors should give detailed information of the network in a table.
2. Effective comparative experiments are lacking in the paper.

---

## Round 0.2 · Minor Revisions

Dear Authors,

Your paper has been revised. Based on the reviewers' reports, it needs revisions before being accepted for publication in PeerJ Computer Science. The main issue is related to the lack of explanations on how to implement the numerical examples discussed in your paper. A flow diagram of your rationale for conducting your simulations should be helpful to this end.

Reviewer 1 ·

Basic reporting

The Authors have addressed all issues and comments from the previous round.

Experimental design

The Authors have addressed all issues and comments from the previous round.

Validity of the findings

The Authors have addressed all issues and comments from the previous round.

Additional comments

The Authors have addressed all issues and comments from the previous round.

Reviewer 2 ·

Basic reporting

This article introduced a convolutional attention network method for detecting HVCB mechanical faults. But the following problems still exist, and the author is suggested to revise them carefully.
1. The article mentions the application of a convolutional attention networks for fault detection of circuit breakers, but it does not introduce this theoretical method, let alone explain how to apply this method for fault detection. This article is not like a scientific paper, but more like an introductory explanatory article.
2. Unfortunately, I did not see significant contributions from the current Sections material & methods.

Experimental design

In section experiments of the article, some simulation examples are provided, but it does not provide how to implement them using the research methods mentioned in the article? Please provide detailed explanations.

Validity of the findings

no comment

---

## Round 0.3 · accepted · Accept

Dear Authors,
Your paper has been revised. Based on the reviewers' reports it can be accepted for publication in PEERJ Computer Science.

Thank you for your fine contribution.

Reviewer 1 ·

Basic reporting

The Authors have addressed all issues and comments from the previous round.

Experimental design

The Authors have addressed all issues and comments from the previous round.

Validity of the findings

The Authors have addressed all issues and comments from the previous round.

Additional comments

The Authors have addressed all issues and comments from the previous round.

Reviewer 2 ·

Basic reporting

Basically meets the requirements

Experimental design

Basically reasonable

Validity of the findings

Basically reasonable

Additional comments

Some issues have been modified and explained, but their scientific validity is not high. I will reserve my opinion.

Annotated reviews are not available for download in order to protect the identity of reviewers who chose to remain anonymous.

Reviewer 3 ·

Basic reporting

no comment

Experimental design

no comment

Validity of the findings

no comment

Additional comments

The authors have revised the questions proposed, but there are still some issues that need to be improved.
1 There is no summary of the existing problems in the introduction.
2.There is a lack of validation of the results by ablation studies in the comparison of methods thus failing to support some of the conclusions drawn by the authors. For example, validation of attention mechanisms, validation of multi-sensor data.